# Circulating Tumor Cells: Technologies and Their Clinical Potential in Cancer Metastasis

**DOI:** 10.3390/biomedicines9091111

**Published:** 2021-08-30

**Authors:** Jerry Xiao, Paula R. Pohlmann, Claudine Isaacs, Benjamin A. Weinberg, Aiwu R. He, Richard Schlegel, Seema Agarwal

**Affiliations:** 1Tumor Biology Training Program, Lombardi Comprehensive Cancer Center, Georgetown University, Washington, DC 20007, USA; jx109@georgetown.edu; 2Department of Breast Medical Oncology, The University of Texas MD Anderson Cancer Center, Houston, TX 77030, USA; prpohlmann@mdanderson.org; 3Center for the Study of Sex Differences in Health, Aging & Disease, Lombardi Comprehensive Cancer Center, Georgetown University Medical Center, Washington, DC 20007, USA; isaacsc@georgetown.edu; 4Ruesch Center for the Cure of Gastrointestinal Cancers, Lombardi Comprehensive Cancer Center, Georgetown University Medical Center, Washington, DC 20007, USA; Benjamin.A.Weinberg@gunet.georgetown.edu (B.A.W.); Aiwu.R.He@gunet.georgetown.edu (A.R.H.); 5Center for Cell Reprogramming, Department of Pathology, Georgetown University Medical Center, Washington, DC 20007, USA; schleger@georgetown.edu

**Keywords:** circulating tumor cells, tissue culture, CTC-derived xenografts, metastasis, clinical trials, cancer, drug screens

## Abstract

Circulating tumor cells (CTCs) are single cells or clusters of cells within the circulatory system of a cancer patient. While most CTCs will perish, a small proportion will proceed to colonize the metastatic niche. The clinical importance of CTCs was reaffirmed by the 2008 FDA approval of CellSearch^®^, a platform that could extract EpCAM-positive, CD45-negative cells from whole blood samples. Many further studies have demonstrated the presence of CTCs to stratify patients based on overall and progression-free survival, among other clinical indices. Given their unique role in metastasis, CTCs could also offer a glimpse into the genetic drivers of metastasis. Investigation of CTCs has already led to groundbreaking discoveries such as receptor switching between primary tumors and metastatic nodules in breast cancer, which could greatly affect disease management, as well as CTC-immune cell interactions that enhance colonization. In this review, we will highlight the growing variety of isolation techniques for investigating CTCs. Next, we will provide clinically relevant context for CTCs, discussing key clinical trials involving CTCs. Finally, we will provide insight into the future of CTC studies and some questions that CTCs are primed to answer.

## 1. Circulating Tumor Cells in Cancer Metastasis

Cancer metastasis, the spread of cancer cells from a primary site to distant organs, is cited as the contributing cause of cancer-associated death in two out of every three solid tumors [1]. Yet, despite the heavy clinical burden of metastasis, much remains to be uncovered about this phenomenon. Presently, the metastatic cascade can be broken down into several discrete stages: (1) local invasion, in which cells leave a primary site and move towards the circulation; (2) intravasation, during which cancer cells will activate extracellular matrix (ECM)-altering factors, epithelial-mesenchymal transition (EMT) factors, and other signaling pathways to migrate through vessel endothelium; (3) circulation, during which cancer cells travel throughout the body on the circulatory highway; (4) extravasation, the movement of cells through a distal endothelium; and (5) colonization, the seeding and subsequent outgrowth of tumor cells at a distant site [2].

Scientific advancements in the understanding of metastasis have only added increasing complexity to the metastatic cascade. For example, organotropism—the preferential metastatic seeding of specific organs—is consistently observed in the clinical setting [3]. Seminal work in the early 2000s characterized distinct transcriptomic profiles associated with breast cancer organ-specific metastasis to the brain and lungs, proposing a revolutionary dynamic that altered the view of metastasis to become a discretionary process, rather than a completely random phenomenon [4,5]. Additionally, informed by observations of similar invasive phenotypes between embryogenesis and cancer metastasis, links to the activation of EMT in the earlier invasion and intravasation as well as later extravasation stages of cancer metastasis added a temporal element to the model [6]. More recently, evidence suggests that tumors may pre-emptively prepare distant organs for colonization by metastatic seeds through the formation of pre-metastatic niches [7]. These ideas all point to a similar theme—that what we know about metastasis is far less compared with what is unknown. Notably, most of these discoveries have been reserved for either early- or late-stage metastatic cascade events, with little attention placed on the circulation stage, in which cells travel towards their eventual colonizing sites. This population of traveling cells is known as circulating tumor cells (CTCs) [8].

CTCs are a key intermediary stage in cancer metastasis that are well-suited for clinical study [8]. Specifically, CTCs can be captured and studied in a relatively non-invasive manner through simple blood draws as opposed to tissue biopsies [9]. The ease of collection, in addition to an increasing emphasis on personalized medicine, foreshadows an important role for CTCs in the study of metastasis. In this review, we will begin by introducing the varying methods used for identifying, isolating, and profiling CTCs, as well as current controversies surrounding the identity of CTCs. Next, we will highlight the importance of CTCs in the clinical setting, introducing trials in which CTCs have played a role in monitoring or influencing therapeutic decisions. Finally, we will preview the future of CTCs, with a brief discussion on the next stage of CTC study involving attempts to grow CTCs in the laboratory.

## 2. Technologies for Isolating CTCs

Detection of a population of cancer cells within the circulatory system of cancer patients was first reported in the 1990s by Racila et al., [10] using an immunomagnetic separation and flow cytometry protocol relying on epithelial cell adhesion molecule (*EpCAM*)-positive expression and anti-CD45 and nucleic dye exclusion of contaminant leukocytes and red blood cells (RBCs), respectively. Importantly, in samples from a healthy, non-cancerous patient, there was little to no *EpCAM*-expressing cells in circulation [10]. This *EpCAM*-positive, *CD45*-negative definition would become the basis of the initial definition of CTCs and serve as the foundation for the development of CellSearch, an antibody-dependent device for the enumeration, or counting, of CTCs from whole blood samples of patients [11]. CellSearch^®^ (Menarini Silicon Biosystems Inc., Huntingdon Valley, PA, USA) would become the first and only FDA-approved device to date for CTC enumeration in 2008 [11]. 

While CellSearch^®^ remains the only FDA-approved method for enumeration in the clinic, multiple other devices and principles have been employed by researchers to isolate CTCs. In general, these devices can be classified based on three principles for identifying CTCs from a whole blood sample: (1) antibody-based marker-dependent platforms; (2) secreted proteins and transcriptomic-based platforms; and (3) physical characteristic-based platforms (Figure 1).

### 2.1. Antibody-Based Marker-Dependent Platforms

Antibody-based marker-dependent platforms, such as CellSearch^®^, utilize a cocktail of antibodies targeting various cell surface markers to pull down or eliminate cells (Figure 1A). Importantly, the ability of antibody-based marker-dependent platforms to capture CTCs is highly dependent on the exact cocktail of antibodies used. In theory, patients diagnosed with active metastatic disease should all present with at least a few CTCs within their bloodstream [12]. In practice, this 100% theoretical yield is hardly ever achieved, revealing a limit to the sensitivity of modern CTC isolation methods (Table 1). For example, in one of the earliest reports during its development, CellSearch^®^ yielded CTCs in only 61/177 (37.2%) samples, despite all patients being diagnosed with active metastatic breast cancer (mBC) [8]. Cytosorter, an immunomagnetic-based platform similar to CellSearch^®^, was able to detect *EpCAM*-positive CTCs in 32/36 (88.9%) of mBC samples [13]. Notably, CellSearch^®^, Cytosorter, and other antibody-dependent platforms such as MagSweeper also use antibodies conjugated with ferrous elements to perform magnet-based separation [8,13,14]. 

On the other hand, increasing emphasis has been placed on the development of microfluidic platforms in the form of chips which are relatively more accessible and easier to use than magnetic antibody-based protocols. These chips utilize anchored antibodies detecting CTC surface markers to the chip substrate, immobilizing CTCs while allowing contaminant RBCs and immune cells to flow freely through the chip. For example, a commonly used alternative microfluidic platform is the CTC-chip, which would later be developed into the herringbone chip (HB-chip) based on improvements to the microfluidic channel design [20,43,44]. Both the CTC-chip (23/36, 64%) and HB-chip (14/15, 93%) reported higher isolation success rates from samples obtained from the blood of metastatic pancreatic cancer patients compared with CellSearch^®^ [20,43]. 

CTCs are a highly heterogeneous population of cells, arguing against the narrow isolation criterion set by the initial *EpCAM*-dependent platforms [45]. Interestingly, in the initial evaluation of the 2016 EORTC TREAT-CTC trial, researchers suggested that unexpectedly low CTC counts discovered in the samples were likely due to the reliance on a technology that restricted CTC detection to those cells expressing *EpCAM*, which may in turn have affected the accurate tracking of treatment response [46]. In their discussion, the researchers further suggested that other methods for isolating CTCs using a broader definition of CTCs might be able to overcome this obstacle [46]. 

Concurrent to this trial, other groups were also working to develop platforms that instead captured cells using a cocktail of antibodies rather than a single antibody. For example, a recently reported platform HB-MFP [47] utilizes a similar herringbone design as the HB-chip, but iterates on the platform using antibodies against *EpCAM* in addition to prostate cancer-specific markers *PMSA* and *PSA*. In a proof-of-principle study, HB-MFP isolation captured 6–280 CTCs/mL of blood in all nine stage-2+ localized or metastatic prostate cancer patients tested [47]. However, the capture of CTCs from patients with non-small cell lung cancer (NSCLC) or small cell lung cancer (SCLC) comparing quadcapture (four antibody cocktail) and unicapture method highlights some of the issues with using antibody cocktails for CTC isolation [48]. In this study, quadcapture using antibodies against *MUC-1, EGFR, HER2*, and *EpCAM* resulted in the identification of CTCs in 20% of NSCLC and 80% of SCLC samples, while capture using just a singular anti-*EpCAM* antibody improved NSCLC capture (40%) but decreased SCLC capture (60%) rates [48]. As demonstrated by this study, antibody cocktails, while improving sensitivity, are limited by the selection of appropriate markers. Identifying the correct cocktail of cancer-specific markers to target therefore erects an additional barrier to the adoption of these technologies, as many of these cocktails are not likely to be generalizable for all cancers. Because of this, multiple different platforms would be required, and each would be applicable only to a small subset of patients with cancer, directly influencing technology accessibility and ease of use.

Based on these and other studies, it becomes clear that antibody-based methods for capturing CTCs are inherently limited by the narrow definitions imposed by using cell-surface markers. Evolving definitions of CTC cell surface markers continue to highlight issues with using single-antibody isolation methods. In these cases, not all CTCs may be captured, resulting in loss of potentially important subsets of CTCs for metastasis, as seen by the inability of any platform to achieve the 100% theoretical yield threshold. While circulating *EpCAM*-positive cells represent a solid foundation for the capture of CTCs, an antibody-cocktail—preferably one that is generalizable across multiple cancer types—could represent the most ideal solution for increasing sensitivity of these CTC capture platforms. 

### 2.2. Secreted Proteins and Transcriptomic-Based Platforms

An alternative approach that some platforms utilize is based on the principle that CTCs should exhibit a unique molecular signature. This subset of methods, therefore, seeks to isolate CTCs via the expression or secretion of cancer-specific genes or gene products (Figure 1B and Table 1). Two examples of these isolation platforms are the EPIthelial ImmunoSPOT (EPISPOT) [33,38,49] and AdnaTest (Qiagen, Germantown, MD) [23,26,29,30,50,51,52]. Both EPISPOT and AdnaTest propose that the detection of secreted proteins or transcripts rather than cell surface markers better selects for the proportion of living CTCs that would be more likely to result in metastasis as opposed to the less important population of apoptotic CTCs [53]. 

In a preliminary prospective study examining CTC detection in 254 mBC patients, EPISPOT detection of *CK19* protein identified 115/194 (59%) positive samples whereas CellSearch^®^ identified 122/254 (48%) positive samples [33]. Simultaneous processing of samples with both technologies only agreed on CTC positivity in 76% of samples [33]. Later, the same group also reported similar improvements in CTC detection favoring EPISPOT detection of *S100* protein, a metastatic melanoma marker [38,54]. In a similarly designed study, AdnaTest, which identifies CTCs based on reverse transcriptase-PCR (RT-PCR) amplification of cancer-specific markers such as *EGFR, CEA*, and *EpCAM*, demonstrated increased sensitivity when compared with CellSearch^®^ [50,52]. Importantly, the commercially available AdnaTest begins with an immunomagnetic isolation step for cell surface markers (not identified), making AdnaTest subject to the same limitations that antibody-based platforms would be [50]. Alternatively, the same group also reported a direct detection RT-PCR protocol examining a panel of prostate cancer-specific genes, demonstrating higher sensitivity compared with both CellSearch^®^ and AdnaTest [50]. 

Both AdnaTest and EPISPOT work towards isolating a more functionally relevant population of CTCs. Studies examining CTC apoptosis have revealed that at any time point, 50–80% of CTCs express apoptotic markers, consistent with the theory that not all CTCs will successfully extravasate [53]. Prioritizing the identification of living CTCs more likely to survive in circulation would significantly enhance the clinical relevance of CTC isolation platforms. On the other hand, protein secretion and transcriptomic expression are highly transient processes, which subject secreted protein- and transcriptomic-based platforms to the whims of these processes [55]. Furthermore, as an added wrinkle to transcriptomic processes, another dynamic arises when considering the importance of levels of amplification when compared with healthy tissue [56]. Identifying appropriate cut-offs for considering RT-PCR positivity would require robust and extensive cancer-specific testing. Finally, both EPISPOT and AdnaTest involve the lysing of captured CTCs, which limits their use for downstream analyses such as cell culture and drug screens.

### 2.3. Physical Characteristic-Based Platforms

Both antibody-dependent and transcriptomic/proteomic-based platforms involve the isolation of a defined CTC population. For these platforms to succeed, there needs to be a strong foundation of understanding of the definition of CTCs. Using a too narrow or too broad definition could therefore result in either a loss of important CTCs or a loss of specificity and resulting introduction of contaminant populations such as red blood cells. Dissociating CTC definitions from transient cellular processes such as protein secretion and transcriptomic expression could potentially lead towards higher capture rates of CTCs from whole blood samples by inherently broadening the selection criterion. Additionally, in both antibody- and transcriptomic-/proteomic-based protocols, cells often undergo extensive manipulation or even lysing, rendering them incompatible with further downstream analysis. In response, multiple groups have devoted work towards isolating CTCs based solely on the expected physical properties of these cells (Table 1). In general, the majority of these platforms utilize microfluidic devices to filter out contaminant components of whole blood such as red blood cells (RBCs) and leukocytes (Figure 1C).

One of the easiest to understand principles within this group of platforms involves the isolation of CTCs based exclusively on size [19,31,34,57,58]. Normal human RBCs and WBCs have an average diameter of about 7–9 µm compared with CTCs, with an average size of 30 µm [59]. Early on in the development of size-based platforms, researchers discovered that a common fault involved filter clogging by the high concentration of RBCs compared with CTCs [60]. The majority of these devices, therefore, iterate on themselves by combining multiple filters of varying pore size or combining other principles for CTC separation [19,58,61]. For example, in an early report comparing isolation techniques, researchers identified ≥5 CTCs in 18/22 (82%) samples from pancreatic patients using a size-based trap compared with 9/22 (41%) in the same samples using CellSearch^®^ [34]. Furthermore, the size-based trap resulted in a mean count of 257 CTCs/7.5 mL of whole blood, which was significantly higher than the average CellSearch^®^ count of 25 CTCs/7.5 mL of whole blood, once again suggesting the loss of a significant population of CTCs using only the narrow *EpCAM*-positive definition employed by CellSearch^®^ [34]. Alternatively, other physical characteristics such as cell membrane conductance [62,63], flow parameters [36], cell deformability [64,65], acoustic properties [66], and density [67,68,69] have also been proposed as a method for CTC selection from whole blood. Importantly, some of these isolation methods encourage the capture of not only CTCs but also accompanying cancer immune cells that support CTC viability, better mimicking the heterotypic interactions that occur in vivo [68,70,71,72,73]. 

Ultimately, CTC isolation platforms have diverged based on the principles employed for defining CTCs. Importantly, the development of all platforms involves a careful balance of sensitivity, the capture of all CTCs, and specificity, the capture of only CTCs. A consensus opinion on the appropriate selection criterion for CTCs would best solve this problem. Unfortunately, this is easier said than done.

## 3. CTCs: Epithelial or More?

So far, no platform reported has demonstrated a robust ability to achieve the theoretical 100% positivity in CTC capture expected from metastatic cancer samples (Table 1). Early methods for capturing CTCs relied on a narrow but important distinction between healthy and cancerous cells within circulation—the presence of epithelial cells defined by nucleic acid^+^, cytokeratin^+^, and *CD45*^-^ [8,10]. As previously discussed, new platforms being developed have shifted attention away from simply enumerating CTCs to evaluating CTCs on a transcriptomic and genomic level. As new profiling results have revealed, CTCs are a highly heterogeneous population reaching beyond the simple epithelial definition first reported in the 1990s [10,45,74,75,76,77,78]. Together, these studies argue against the concept of a lone CTC cell acting out to induce metastasis, introducing transcriptomic and proteomic dynamics as well as other non-CTC players such as cancer-associated neutrophils and macrophages, acting together to induce metastasis.

In the first-ever paper reporting single-cell RNA-sequencing (scRNA-seq) of CTCs, Powell and colleagues evaluated 510 mBC CTCs to demonstrate considerable heterogeneity in the expression of metastasis-associated genes such as *NPTN, S100A4, S100A9*, and EMT factors, compared with a panel of breast cancer cell lines [75]. In a separate study, expression of the evaluated factors varied on a cell-by-cell basis, suggesting high transcriptomic heterogeneity among CTCs. Similarly, single-cell proteomics analyses performed on melanoma patients demonstrated considerable cell-by-cell receptor variations that may be directly linked to tumor response [79]. To further add to this story, a dual-color RNA in situ hybridization (ISH) evaluating the expression of a panel of epithelial and mesenchymal transcripts revealed that CTCs exist along a spectrum of EMT [45]. Somewhat surprisingly, a small but not insignificant portion of captured CTCs expressed little to no epithelial markers [45]. Additionally, CTCs existing in clusters, which have been shown to metastasize more prevalently in vivo, tend to have stronger mesenchymal expression compared with epithelial expression [32,45,80]. It follows, therefore, that *EpCAM*-dependent antibody-based protocols for isolating CTCs could miss out on this important mesenchymal population. 

In addition to these findings, this report and others have routinely demonstrated that CTC gene expression often shifts according to patient chemotherapy treatment [45,77,78]. For example, in a seminal work performing longitudinal RNA sequencing on CTCs extracted from metastatic breast cancer patients, researchers revealed not only that CTCs exhibited a spectrum of “EMT” phases based on EMT marker expression, but that the proportion of CTCs within each EMT platform was dramatically altered following patient treatment [45]. Longitudinal profiling of CTCs could therefore give clinicians a pseudo-real time, relatively non-invasive method for evaluating tumor identity.

Finally, several groups performing mutational analyses on captured CTCs have reported evidence of not only transcriptomic variation but also mutational variability among CTCs and discordance between CTCs and primary tumors [76,81,82]. For example, in one of the seminal works of the field, researchers performed array-comparative genomic hybridization and next-generation sequencing to reveal mutational discordance between CTCs and matched patient primary tumors [76]. Specifically, the group reported changes in copy number variation in 39.5% of samples tested in several critical cancer-associated genes that might affect therapeutic treatment, including mutations in *KRAS* and *PIK3CA* [76]. Others have reported 5–50% mutational discordance in oncogenic genes surveyed, highlighting the high heterogeneity that CTCs exhibit [81,82].

These studies and many others have presented clear evidence that CTCs are a heterogeneous population. Ultimately, the considerable heterogeneity of CTCs is a significant barrier preventing the efficient and accurate capture of this promising population. It also becomes increasingly important to be able to accurately model this heterogeneity if CTCs are to be appropriately used for studying metastasis [32,45]. 

## 4. CTC Enumeration and Its Clinical Relevance

While the exact definition and isolation method used to capture CTCs remains a controversial and highly discussed field, ample evidence has already been provided to support the clinical relevance of CTCs. Whether through enumeration or longitudinal monitoring and profiling, CTCs have been shown to serve as a robust independent prognosticator as well as a method for evaluating patient response while on therapy. 

### 4.1. CTCs as an Independent Prognosticator

Evidence that CTC enumeration could be an independent predictor of progression-free survival (PFS) and overall survival (OS) arrived as early as 2004 [8]. In this groundbreaking work, 177 patients with mBC were enrolled and had for CTC counts measured in peripheral blood at baseline and following treatment. Importantly, only 2/345 (0.6%) of healthy individuals or patients diagnosed with benign disease had ≥2 CTCs using CellSearch^®^, while 108/177 (61%) of patients diagnosed with mBC had ≥2 CTCs [8]. Using a cut-off of 5 CTCs/7.5 mL of blood, patients were stratified into a high-CTC (87/177, 49%) or low-CTC group [8]. Notably, the high-CTC group had significantly shorter median PFS and OS compared with individuals with lower CTC counts. Furthermore, a cut-off of 5 CTCs/7.5 mL of blood (approximately one EDTA-vacutainer tube) provides context for the rarity of CTCs in blood, which is often cited to be as little as one CTC per 10^9^ hematologic cells in blood [10,11].

Following the approval and commercialization of CellSearch^®^ in 2008, numerous studies spanning a wide range of cancer types, including breast [11,15,16,22,25,26,30,32,33,39,40], colorectal [17,22,27], pancreatic [22], prostate [18,20,21,25,32,50,83], lung [24,28], melanoma [38], bladder [41,84], and gynecological [37] cancers have all confirmed a similar role for CTC enumeration within the clinical setting. The utility of CTC enumeration in stratifying patient prognosis may best be punctuated by a 2019 international expert consensus paper analyzing individual patient data from 18 cohorts encompassing 2436 mBC patients, confirming that stage IV mBC patients could be stratified into stage IV_indolent_ and stage IV_aggressive_ subtypes based solely on CTC count, independent of other clinical and molecular variables [39]. 

### 4.2. CTCs in Clinical Therapy

In addition to being able to predict patient prognosis, several clinical trials have demonstrated the usefulness of CTCs in monitoring or guiding clinical therapy [46,85,86,87,88,89,90,91,92,93,94,95] (Table 2). For example, longitudinal studies evaluating CTC status of patients following ALK inhibitor treatment revealed elimination of CTCs after treatment could predict disease remission [85]. Similarly, in NSCLC and [86] and castration-resistant prostate cancer [91], treatment resulted in decreased CTC counts associated with longer PFS in both studies. In another study, longitudinal monitoring of CTC counts in colorectal cancer preceded clinical symptoms and radiological imaging in a patient diagnosed with local tumor recurrence thirteen months after tumor resection [42]. Furthermore, in the phase III SWOG S0421 trial examining docetaxel-treated castration-resistant prostate cancer, CTC counts were significantly associated with baseline PSA, bone pain, liver disease, hemoglobin, alkaline phosphatase, and PSA and response evaluation criteria in solid tumors (RECIST) response, putting forth evidence that CTC counts could be used as a viable alternative index in determining treatment [91]. Recently, another prospective trial revealed patients presenting with high androgen-receptor-related gene expressing CTCs were more likely to present with castration-resistant versus castration-sensitive prostate cancer as well as significantly shorter overall survival [96]. Interestingly, others have shown that CTC numbers can increase dramatically in the perioperative period following surgery, highlighting the real-time sensitivity of CTC enumeration for monitoring disease progression [97].

In the only reported study of CTC-guided management, the 2021 STIC-CTC trial compared CTC-guided therapy against investigator choice for the management of hormone receptor-positive, HER2-negative mBC [87]. Out of 755 patients enrolled, 377 patients were randomly allocated to the CTC arm [87]. Using a cutoff of 5 CTCs/7.5 mL of blood, patients were treated with either endocrine therapy (<5 CTCs/7.5 mL) or chemotherapy [87]. Patients enrolled in the CTC count-driven arm had a greater median PFS compared with the investigator-driven arm, suggesting a benefit of using CTC-guided treatment algorithms [87]. In the future, studies could use either CTC count or CTC gene expression (i.e., biomarker expression levels greater than a certain amount over healthy tissue) as determining factors for treatment. 

Remarkably, one interesting trend that would significantly affect clinical management of metastatic breast cancer has been observed throughout multiple CTC-involved clinical trials such as the GEPARQuattro [88], DETECT III [90], EORTC TREAT-CTC [46,93], and SUCCESS-B [89] clinical trials. To begin with, researchers frequently observed that treatment regimens in these studies were often ineffective in producing a tumor response. Attempting to determine an answer to the low tumor response, the various trials consistently observed receptor status switching between CTCs derived from metastatic breast cancer samples and primary tumors [23,88,89,90]. For example, in one multi-center prospective trial, researchers discovered only a 50% concordance of HER2 expression between CTCs and primary tumors across 254 patients studied [23]. Discordant expression of receptors between captured CTCs and primary tumor was prevalent throughout many patients, suggesting a critical potential limitation of using single-agent targeted therapy when treating patients with metastatic cancer [46,88,93]. Overall, these trials indicate the importance of evaluating CTCs and the critical information provided by proper characterization of this unique population.

Based on these studies, enumeration of CTCs alone already provides a significant and easily accessible clinical tool that is not provided by invasive tissue biopsies. Longitudinal blood draws following treatment have also been shown to be a robust index for monitoring tumor response to treatment [86,90,98]. In the future, enumeration of CTCs through blood draws at earlier stages of cancer could potentially improve clinical prediction of cancer metastasis and progression, thereby providing clinicians with a powerful tool for monitoring cancer.

## 5. Growing CTCs Ex Vivo: The Next Frontier

CTC enumeration has become a well-established clinical tool that can be used for monitoring therapeutic response and predicting patient prognosis. As CTC isolation platforms are continuously iterated to improve capture efficiency and accuracy, there has concurrently been a shift in priorities towards extracting live, viable CTCs. By extracting and growing CTCs ex vivo, researchers could not only generate new patient-derived models for personalized medicine but also generate additional source material for more powerful, next-generation molecular techniques, enabling multi-omics level study of CTCs. Currently, the two predominant principles used to propagate CTCs involve (1) the growth of CTCs in tissue culture or (2) the direct injection of CTCs into immunocompromised mice, forming CTC-derived xenografts (CDXs). Discussion of the specifics of these methods falls out of the scope of this review, but is covered in-depth by others [99,100].

The ability to expand CTCs ex vivo would be a critical tool for the cancer researcher to have access to. By expanding CTCs, researchers can begin capitalizing on the promised potential of CTCs. Several significant contributions have already arisen from CTC cultures. For example, CTC co-cultures with cancer-associated fibroblasts (CAFs) and cancer-associated neutrophils/macrophages (CANs/CAMs) have given researchers the ability to investigate the heterotypic interactions supporting CTCs within the bloodstream [68,101,102]. CTC cultures or CDXs have also been leveraged to perform drug screens to test new therapeutic compounds targeting metastatic cell populations [103,104,105]. For example, in several CDX models, treatment with chemotherapy agents mirrored or predicted patient response to treatment [103,104,105]. Depending on the time frame established, CTC cultures and CDX models, combined with the emphasis on personalized medicine, could be used to inform clinical decisions and select appropriate therapeutic regimens [68]. 

Unfortunately, a limiting factor of CTC culture and CDX models remains the absence of highly efficient methods. Most reported methods rely on rare circumstances involving high CTC counts in order to achieve success [101,106,107,108,109,110]. However, recent advances in culturing methods by our lab have reported a high success rate in establishing CTC cultures from 12/12 metastatic breast cancer samples [68]. Furthermore, reported CTC cultures often have significantly long CTC doubling times, which may affect their usability in certain, highly aggressive cancers during which patient prognosis is short [110,111]. Similarly, a nagging issue for CDX models is their long time for follow-up, limiting their clinical utility [104,112,113,114,115]. Overall, there have been suggestions that CTCs may be hampered by slow proliferation due to the cell’s opposing migratory and proliferative states. Specifically, the cancer cell’s urge to migrate through the circulatory system versus the cells interest in proliferating are often regulated by different and sometimes opposing signaling pathways [116,117]. Finally, there is currently minimal evidence of whether successful CTC cultures accurately model the heterogeneity of CTCs in vivo, while CDX models have reported conflicting results regarding metastatic behavior, a troubling phenomenon that requires further investigation [104,110,111,112,113,114,115]. Despite these limitations, these initial steps towards propagating CTCs ex vivo are part of a promising movement towards unlocking the potential of CTCs.

## 6. Future of CTCs in Personalized Medicine

The studies reviewed here have demonstrated the importance of studying CTCs—from independently predicting patient prognosis to revealing hormone receptor heterogeneity to identifying mutational discordance—which can work together to improve the quality of care in the clinic. Recent advancements in the understanding of CTCs have resulted in changes to the definitions used for isolation and a movement towards unbiased, antibody-independent methods for capturing a more accurate representation of the heterogeneous population. Once expanded, functional studies using CTCs could help researchers guide therapeutic decisions and make metastasis-specific therapeutic contributions (Figure 2). In the future, expanded CTCs could be used for high-throughput compound testing to guide the development of metastasis-preventing drugs as opposed to currently available metastasis-mitigating drugs. Unfortunately, methods for expanding CTCs in the laboratory and in vivo in CTC-derived xenografts have been met with low success rates, with success oftentimes reserved only for the rarest samples that have high CTC counts at baseline [68]. We theorize that patient-derived CTCs could be eventually propagated using a CTC culture prior to CDX injection, improving chances for CDX tumor formation and enabling extended molecular interrogation of CTCs, with or without the requirement of a CDX model.

Additionally, early findings from clinical trials such as the STIC-CTC trial suggest a promising role for CTCs in personalized medicine [87]. Future studies may benefit from evaluating CTC-guided therapy, either through CTC count or CTC profiling, as an added index for consideration in treatment algorithms. As multiple clinical trials have already shown, CTC counts can retrospectively be used to measure treatment response. It would therefore not come as a surprise that CTC count could be used prospectively to guide treatment decisions. 

Increasing reports of next-generation sequencing of CTCs and CDXs also suggest a place for CTCs in studying metastasis. CDX models have already commonly been used to perform in vivo drug screens to high success [103,104,105]. Harvesting and sequencing of matched CDX primary tumors and metastatic nodules could also highlight genetic drivers involved with organ-specific metastasis [104,112,114]. Alternatively, the identification of enriched pathways and biological phenomena in successful CDXs provides insight regarding potential targets for metastasis-preventing therapy. However, although useful for enhancing the knowledge of cancer development and aiding cancer therapy development, CDX models may not advance as a routine clinical strategy to personalize treatment to every cancer patient given its complexity and lengthiness [118]. Finally, as technologies for isolating CTCs continue to become more efficient, CTC capture from early-stage cancer patients could be paired with CDX models to further increase the benefits of CTC study by enabling early detection of metastatic cells.

Finally, integration of CTC characterization with other components of the liquid biopsy, such as matched whole blood genomic DNA, cell-free DNA, and extracellular vesicle mRNA, could improve the prognostic and clinical relevance of CTCs [119]. While discussion of the relevance of other components of a liquid biopsy is not within the scope of this review, several promising studies have already confirmed a similar clinical utility from examining these products as CTCs [119]. In the future, clinicians may utilize all aspects of the liquid biopsy to supplement traditional measurable indices from the blood and tissue biopsies to develop a more accurate snapshot of the disease status of a patient. With all of this information in hand, more informed decisions by the clinician will likely translate to better standards of care and improved quality of life for many cancer patients.

## 7. Conclusions

Existing studies of circulating tumor cells have already reported high-impact findings that substantially alter our understanding of metastasis. CTCs have the advantage of liquid biopsy techniques, precluding the need for a true tissue biopsy which may not be technically feasible or repeated multiple times for certain patients. In addition, CTCs are a lively fraction of the tumor, in contrast to ctDNA. However, CTCs tend to occur in low numbers, currently requiring single-cell resolution techniques that may be prone to random variation and low coverage, potentially misleading conclusions that should inform patient care decisions. For clinical utility, it is critical that relevant clones are represented and in numbers large enough to allow unbiased analysis that could in fact have an impact on patient care.

Prioritization of the development of efficient isolation and propagation technologies would further improve the utility of CTCs in the clinical setting. Expansion of these studies towards early-stage cancer patients could additionally increase the pool of patients that could benefit from CTC research. Profiling of CTCs and CDX models could subsequently help lift the shroud of mystery surrounding cancer metastasis. Since a large majority of cancer-associated death can be attributed to the burdens brought on by metastasis, it is imperative that research efforts turn towards answering these questions. Finally, due to the relatively non-invasive methods required for their study, CTCs are primed to become a crucial and informative component in the future of personalized medicine.

## Figures and Tables

**Figure 1 biomedicines-09-01111-f001:**
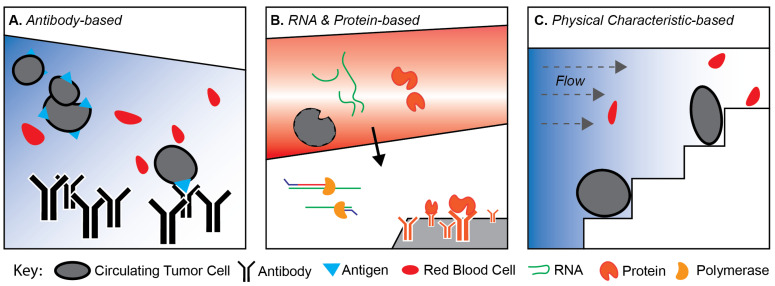
Platforms for circulating tumor cell isolation can be split into three main principles: (**A**) antibody-based, which utilizes antibodies to capture cell surface markers commonly expressed on circulating tumor cells (e.g., EpCAM), (**B**) RNA and protein-based, which detect the presence of CTCs based on either expressed RNA and/or secreted proteins, and (**C**) physical characteristic-based protocols, which capture CTCs from whole blood based on anticipated physical differences in CTCs and other cell populations such as red blood cells.

**Figure 2 biomedicines-09-01111-f002:**
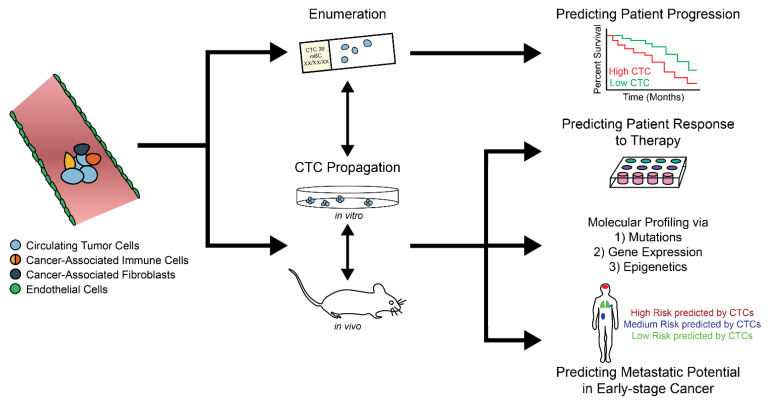
Potential applications of circulating tumor cell study.

**Table 1 biomedicines-09-01111-t001:** CTC enumeration studies.

Paper	Platform Type	Cancer Type	CTC-Positivity Rate	Positivity Criteria
Racila et al., 1998 [10]	Antibody	Breast Prostate	29/30 3/3	≥1 CTC ≥1 CTC
Cristofanilli et al., 2004 [8]	Antibody	Breast	108/177	≥2 CTC
Cristofanilli et al., 2005 [15]	Antibody	Breast	43/83	≥5 CTC
De Bono et al., 2008 [11]	Antibody	Breast	125/231	≥5 CTC
Dawood et al., 2008 [16]	Antibody	Breast	114/185	≥5 CTC
Cohen et al., 2008 [17]	Antibody	Colorectal	111/430	≥3 CTC
Scher et al., 2009 [18]	Antibody	Prostate	85/156	≥5 CTC
Tan et al., 2010 [19]	Size Exclusion	Lung	5/5	≥1 CTC
Stott et al., 2010 [20]	Antibody	Prostate	14/15	≥1 CTC
Stott et al., 2010 [21]	Antibody	Prostate	23/36	≥1 CTC
Miller et al., 2010 [22]	Antibody	Breast Colorectal Prostate	125/177 196/413 169/218	≥1 CTC ≥1 CTC ≥1 CTC
Fehm et al., 2010 [23]	Antibody RNA expression	Breast	122/245 90/229	≥5 CTC ≥1 Gene
Krebs et al., 2011 [24]	Antibody	Lung	39/107	≥2 CTC
Armstrong et al., 2011 [25]	Antibody	Prostate Breast	36/38 11/16	≥1 CTC ≥1 CTC
Muller et al., 2012 [26]	Antibody	Breast	116/221	≥5 CTC
Pantel et al., 2012 [27]	Protein expression Antibody	Colorectal	10/53 6/53	≥1 Protein ≥5 CTC
Hou et al., 2012 [28]	Antibody	SCLC	77/97	≥5 CTC
Kasimir-Bauer et al., 2012 [29]	RNA expression	Breast	97/502	≥1 Gene
Strati et al., 2013 [30]	RNA expression	Breast	42/254	≥1 Gene
Hou et al., 2013 [31]	Size Exclusion	Lung	20/20	≥1 CTC
Aceto et al., 2014 [32]	Antibody	Breast	54/79	≥1 CTC
Ramirez et al., 2014 [33]	Protein expression Antibody	Breast	115/194 122/254	≥1 Protein ≥1 CTC
Qin et al., 2015 [34]	Size Exclusion Antibody	Prostate	18/22 9/22	≥5 CTC ≥5 CTC
Danila et al., 2016 [35]	RNA expression	Prostate	34/55	≥1 Gene
Chen et al., 2017 [36]	Cell Flow	Breast Lung	4/4 9/9	≥1 CTC ≥1 CTC
Zhang et al., 2018 [37]	Antibody	Ovarian	98/109	≥2 CTC
Cayrefourcq et al., 2019 [38]	Protein expression Antibody	Melanoma	15/34 10/44	≥1 Protein ≥2 CTC
Cristofanilli et al., 2019 [39]	Antibody	Breast	911/1944	≥5 CTC
Radovich et al., 2020 [40]	Antibody	Breast	50/123	≥5 CTC
Fu et al., 2021 [41]	Antibody	Bladder	?/48	≥1 CTC
Hendricks et al., 2021 [42]	Antibody RNA expression	Colorectal	16/44 33/41	≥1 CTC ≥1 Gene

**Table 2 biomedicines-09-01111-t002:** CTCs in Clinical Disease Management.

Study	Study Population	Study Treatment	CTC Measurement	Results
Nemunaitis et al., 2009 [92]	Advanced NSCLC	Belagenpumatucel-L	CTC enumeration every 4 weeks	Median OS was significantly shorter in patients with ≥2 CTCs/7.5 mL
Riethdorf et al., 2010 [88]	Metastatic breast cancer	Neoadjuvant therapy	CTC enumeration	No association between tumor response and CTC detection
Punnoose et al., 2012 [86]	Advanced NSCLC	Erlotinib + Pertuzumab	CTC enumeration, EGFR expression in CTCs, oncogenic mutations in CTCs	Higher baseline CTC counts were associated with treatment response Larger CTC count decreases at days 14, 28, 56 associated with treatment response
Goldkorn et al., 2014 [91]	Metastatic castration-resistant Prostate Cancer	Docetaxel + Prednisone with or without Atrasentan	CTC Enumeration at baseline and day 21 post treatment	Median OS was significantly shorter in patients with ≥5 CTCs/7.5 mL of blood at day 0 Rising CTC counts from day 0 to day 21 were associated with shorter OS
Agelaki et al., 2015 [90]	Metastatic breast cancer + HER2-positive CTCs	Lapatinib	Immunofluorescent Microscopy stained for HER2/EGFR/Cytokeratin	“Effective in decreasing HER2-positive CTCs… irrespective of HER2 status”
Ignatiadis et al., 2018 [46,93]	High risk, HER2 nonamplified, early breast cancer	Trastuzumab	CTC enumeration at baseline and week 18	Trastuzumab did not decrease rate of CTC detection
Tan et al., 2018 [94]	Colorectal	Chemotherapy (broad)	CTC enumeration	CTC count trends can be used in place of uninformative CEA for monitoring patient response
Bidard et al., 2021 [87]	Hormone receptor-positive, ERBB2-negative metastatic breast cancer	CTC count-driven vs. Clinician-driven first line therapy	CTC enumeration (chemotherapy if ≥5 CTCs, endocrine therapy otherwise)	Median PFS was slightly longer in CTC-driven treatment compared with clinician-driven treatment
Bonvini et al., 2021 [85]	Inflammatory myofibroblastic tumor	Entrectinib	Longitudinal CTC enumeration during treatment (up to 24 months post treatment)	Antitumor activity was associated with elimination of CTCs from blood
Sperger et al., 2021 [96]	Metastatic prostate cancer	Enzalutamide or abiraterone	CTC androgen-receptor (AR) gene expression	CTCs clustered according to AR-gene expression Presence of CTCs enriched for AR-gene transcripts associated with shorter OS
Pang et al., 2021 [97]	Metastatic Breast Cancer	Surgery or Adjuvant Therapy	CTC enumeration	CTC counts will increase in the perioperative period for up to 14 days CTC positivity associated with shorter OS and PFS following adjuvant therapy

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
