# Peer review of "Circulating Tumor Cells: Technologies and Their Clinical Potential in Cancer Metastasis"

_biomedicines, 2021, doi:10.3390/biomedicines9091111_

Round 1
Reviewer 1 Report
This is a well written and interesting review on TTCs that highlights the growing variety of modern TTCs isolating techniques and their clinical potential in cancer metastasis. The topic is promising, given that TCCs exhibit metastatic tropism and reveal genetic drivers of metastasis, features that make them well-suited models for relevant clinical studies, as well as important key players in monitoring and influencing therapeutic decisions for solid malignancies. In addition, in the context of precision and personalized medicine, TTCs have the potential to serve as a readily accessible, transparent window into an individual’s disease.
Technically, the flow and coverage of the several sub-topics addressed in the review are well developed, references are up to date and language editing is not required. The content is aligned with the aims and scope of the journal. Overall, I believe the review is ready to go as it is, without changes or additions.
Author Response
Reviewer #1
Comments and Suggestions for Authors
This is a well written and interesting review on TTCs that highlights the growing variety of modern TTCs isolating techniques and their clinical potential in cancer metastasis. The topic is promising, given that TCCs exhibit metastatic tropism and reveal genetic drivers of metastasis, features that make them well-suited models for relevant clinical studies, as well as important key players in monitoring and influencing therapeutic decisions for solid malignancies. In addition, in the context of precision and personalized medicine, TTCs have the potential to serve as a readily accessible, transparent window into an individual’s disease.
Technically, the flow and coverage of the several sub-topics addressed in the review are well developed, references are up to date and language editing is not required. The content is aligned with the aims and scope of the journal. Overall, I believe the review is ready to go as it is, without changes or additions.
We thank the reviewer for the positive feedback and consideration of this manuscript.

Reviewer 2 Report
Review of manuscript biomedicines-1342439 titled “Circulating Tumor Cells: Technologies and Their Clinical Potential in Cancer Metastasis”
In this review, Xiao et al. are discussing the recent advances in our understanding of the importance of circulating tumor cells (CTCs) in diagnosis and prognosis of cancers. The authors do a phenomenal job of describing the advantages and disadvantages of different technologies used in identifying and isolating CTCs, how these technologies guide clinical practice. The recount of recent clinical trials focusing on CTCs is timely. The authors also provide an insightful discussion of current challenges in the CTC field and what future endeavors should be taken to further the field. Overall, the review is very well written, and warrants publication in Biomedicines with the following minor changes:
- Line 25 – consider changing “the authors” to “we” to achieve a consistent tone throughout the manuscript. The rest of the manuscript refers to the authors as “we”
- Spell out EpCAM the first time it is introduced in the manuscript.
- Add a key to Figure 1 showing what each shape and color in the images refer to (e.g tumor cell/CTC, Antibodies, red blood cells, RNA molecule etc)
- The written figure legend for Figure 1 would benefit from short description of each methodology. The authors should also consider dividing the figure into 3 panels (A through C) and refer to the appropriate sub-panel in the manuscript when each different technology is being discussed.
- The statement in lines 307-312 seems to be missing a citation.
- The authors discuss how CTCs are difficult to grow because of their slow growing nature. A suggesting would be to add a short statement or discussion about why these cells grow slowly (perhaps the opposing cellular states required of movement vs. proliferation).
Author Response
Reviewer #2
Comments and Suggestions for Authors
Review of manuscript biomedicines-1342439 titled “Circulating Tumor Cells: Technologies and Their Clinical Potential in Cancer Metastasis”
In this review, Xiao et al. are discussing the recent advances in our understanding of the importance of circulating tumor cells (CTCs) in diagnosis and prognosis of cancers. The authors do a phenomenal job of describing the advantages and disadvantages of different technologies used in identifying and isolating CTCs, how these technologies guide clinical practice. The recount of recent clinical trials focusing on CTCs is timely. The authors also provide an insightful discussion of current challenges in the CTC field and what future endeavors should be taken to further the field. Overall, the review is very well written, and warrants publication in Biomedicines with the following minor changes:
- Line 25 – consider changing “the authors” to “we” to achieve a consistent tone throughout the manuscript. The rest of the manuscript refers to the authors as “we”
- We appreciate the reviewer’s suggestion and have made the suggested change to Line 25 accordingly.
- Spell out EpCAM the first time it is introduced in the manuscript.
- Epithelial cell adhesion molecule (EpCAM) is first mentioned in line 75, and is elucidated prior to its first mention.
- Add a key to Figure 1 showing what each shape and color in the images refer to (e.g tumor cell/CTC, Antibodies, red blood cells, RNA molecule etc)
- We thank the reviewer’s suggestions and agree that adding a key to Figure 1 would present a clearer figure for interpretation. A key identifying CTCs, antibodies and antigens, RBCs, and RNA and Proteins has been included below the original three panels.
- The written figure legend for Figure 1 would benefit from short description of each methodology. The authors should also consider dividing the figure into 3 panels (A through C) and refer to the appropriate sub-panel in the manuscript when each different technology is being discussed.
- The three panels in Figure 1 have been further subdivided using A through C. Additional callbacks to these respective panels have been included in the text (line 99, 164, and 211). Additionally, brief explanatory details regarding the panels have been included in the figure legend for Figure 1, per the reviewer’s suggestions.
- The statement in lines 307-312 seems to be missing a citation.
- We apologize for this missing citation, and have now included the appropriate citation as reference 71: “Cristofanilli, M.; Pierga, J.-Y.; Reuben, J.; Rademaker, A.; Davis, A.A.; Peeters, D.J.; Fehm, T.; Nolé, F.; Gisbert-Criado, R.; Mavroudis, D.; et al. The Clinical Use of Circulating Tumor Cells (CTCs) Enumeration for Staging of Metastatic Breast Cancer (MBC): International Expert Consensus Paper. Crit Rev Oncol Hemat 2019, 134, 39–45, doi:10.1016/j.critrevonc.2018.12.004.”
- As part of this change, all references within the document were revised to ensure accurate numbering and style for Biomedicines.
- The authors discuss how CTCs are difficult to grow because of their slow growing nature. A suggesting would be to add a short statement or discussion about why these cells grow slowly (perhaps the opposing cellular states required of movement vs. proliferation).
- The reviewer is alluding to evidence that signaling pathways involved in cell migration and invasion are often conflicting with the cell’s interest in proliferating [1,2]. This is certainly a potential reason for the slower doubling times observed in CTC cultures and CDX models. Two sentences putting forth this explanation have been added (Line 403-407), along with the cited papers discussing the detailed mechanisms behind the dichotomy between proliferation and invasion in cancer.
- Gao, C.-F.; Xie, Q.; Su, Y.-L.; Koeman, J.; Khoo, S.K.; Gustafson, M.; Knudsen, B.S.; Hay, R.; Shinomiya, N.; Woude, G.F.V. Proliferation and Invasion: Plasticity in Tumor Cells. P Natl Acad Sci Usa 2005, 102, 10528–10533, doi:10.1073/pnas.0504367102.
- Kohrman, A.Q.; Matus, D.Q. Divide or Conquer: Cell Cycle Regulation of Invasive Behavior. Trends Cell Biol2017, 27, 12–25, doi:10.1016/j.tcb.2016.08.003.
